# A Short-Term Hybrid Energy System Robust Optimization Model for Regional Electric-Power Capacity Development Planning under Different Pollutant Control Pressures

**Jixian Cui [1,2], Chenghao Liao [1,2], Ling Ji [3], Yulei Xie [4,*], Yangping Yu [4] and Jianguang Yin [5]**

[1] Guangdong Provincial Academy of Environmental Science, Guangzhou 510045, China;
cuijx89@hotmail.com (J.C.); liaochenghao@foxmail.com (C.L.)

[2] Guangdong-Hongkong-Macau Joint Laboratory of Collaborative Innovation for Environmental Quality,
Guangzhou 510045, China

[3] School of Economics and Management, Beijing University of Technology, Beijing 100124, China;
hdjiling@126.com

[4] Key Laboratory for City Cluster Environmental Safety and Green Development of the Ministry of Education,
Institute of Environmental and Ecological Engineering, Guangdong University of Technology,
Guangzhou 510006, China; yuyangping@126.com

[5] State Grid Shandong Electric Power Research institute, Jinan 250000, China; 18615683860@163.com

\* Correspondence: xieyulei@gdut.edu.cn

**Abstract:** This paper is aimed at proposing a short-term hybrid energy system robust optimization model for regional energy system planning and air pollution mitigation based on the inexact multi-stage stochastic integer programming and conditional value-at-risk method through a case study in Shandong Province, China. Six power conversion technologies (i.e., coal-fired power, hydropower, photovoltaic power, wind power, biomass power, and nuclear power) and power demand sectors (agriculture, industry, building industry, transportation, business, and residential department) were considered in the proposed model. The optimized electricity generation, capacity expansion schemes, and economic risks were selected to analyze nine defined scenarios. Results revealed that electricity generations of clean and new power had obvious increasing risks and were key considerations of establishing additional capacities to meet the rising social demands. Moreover, the levels of pollutants mitigation and risk-aversion had a significant influence on different power generation schemes and on the total system cost. In addition, the optimization method developed in this paper could effectively address uncertainties expressed as probability distributions and interval values, and could avoid the system risk in energy system planning problems. The proposed optimization model could be valuable for supporting the adjustment or justification of air pollution mitigation management and electric power planning schemes in Shandong, as well as in other regions of China.

**Keywords:** electric power capacity planning; stochastic programming; multi-scenario analysis; emission reduction; system risk aversion

## 1. Introduction

Due to economic development and resource service loads increasing, energy–environmental contradictions (e.g., single-energy structure, small proportion of clean energy, and environmental quality improvement) from energy activities have become a more significantly critical and complex issue in China [1–4]. Although renewable power technology has a widespread application with the national new energy law enacted, China still has a single-energy structure. Additionally, coal-fired power conversion technologies have occupied large proportions of electricity production compared to other energy conversion technologies. According to the China Statistical Yearbook (2018), the amount of electricity generation of coal-fired power has increased from $3.3 \times 10^6$ GWh to $4.4 \times 10^6$ GWh from 2010 to 2016. Moreover, the rapid growth of the electric power industry could pose threats

to environment protection and generate both a series of air pollutants (e.g., sulfur dioxide ($SO_2$), nitrogen oxides (NOx), and particulate matter (PM)) and greenhouse gases [5,6]. The total discharged amount of pollutants from electricity generation has brought significant impacts on the atmospheric environment quality protection, with the values of 1.7 million tonne, 1.6 million tonne, and 0.4 million tonne for $SO_2$, NOx, and PM in 2016, respectively. However, in most energy system planning activities, these environmental effects and their interactions with both energy development and utilization are often ambiguous or uncertain [7–10]. Therefore, a reasonable and effective energy system planning model is important and necessary in balancing regional energy and environmental system sustainable development.

Previously, a number of studies were conducted for planning energy systems and managing air pollution reduction at regional and national scales [11–13]. In real-world management problems, uncertainties exist in many system parameters and within their interrelationships, which could be presented in terms of multiple formats (e.g., interval numbers, probability distributions, and system dynamics) [4,12,14]. Many studies were proposed for energy system optimization and air pollution mitigation models under uncertainty conditions by interval-parameter programming (IPP), fuzzy mathematical programming (FMP), and stochastic mathematical programming (SMP) technologies [15–18]. For example, Dong et al. [19] presented an inexact optimization modeling approach to effectively analyze and address the complexities and uncertainties in energy systems and regarding air pollution mitigation. Zhu et al. [20] advanced an inexact mixed-integer fractional approach for addressing uncertainties and for optimizing management efficiency in sustainable energy systems. In addition, more studies provided multi-stage stochastic programming (MSP) addressing uncertainties and the dynamic performance within energy systems.

According to previous research studies, MSP is used to efficiently address probabilistic uncertainties in the model's right-hand side, which is known as the probability distributions within the multi-stage context [21]. Moreover, MSP has a dynamic characteristic, especially regarding the dynamic transmission of large systems, such as the transfer of capacity expansion in energy systems. However, MSP can hardly address the independent uncertainties of the model's left-hand sides and neither the cost coefficients, and it cannot effectively reduce or avoid system risk, especially regarding the optimization problem, without considering the risk of deviating from the expected value. Risk management entails the exercise of control over some statistical characteristic of the uncertain portfolio return [1,18]. The value at risk (VaR) and conditional value at risk (CVaR) could be used to avoid portfolios that may likely be susceptible to severe losses, which are widely accepted as risk measures in risk management [22–24]. In comparison with VaR, CVaR is a coherent risk measure that has many attractive properties and mainly involves the $\alpha$-quantile and conditional expectation. Through being coupled with CVaR, the conditional expectation of the portfolio returns below a prespecified low percentile of the distribution and the expected losses in severe circumstances can be effectively reflected in large system management problems.

The purpose of this study was to develop an energy system optimization model that combines the interval multi-stage stochastic programming and conditional value-at-risk (CVaR) measure for addressing uncertainties and complexities in the energy system optimization and planning. Focused on the top-level design for the national energy structure adjustment and the environmental protection request, the energy system optimization model was derived from deep deconstruction of regional electric power systems in Shandong Province, as presented in Section 2. Based on energy resource supply, technical processing, demand activities, economic cost/benefits, and the associated air pollution emissions, an energy system optimization model for organizing the relationship was proposed to reflect the dynamics of capacity expansion issues and to address both the uncertain information and system risk that are introduced by the system random characters, as discussed in Section 3. Finally, multi-scenario analysis for electric power system management

in Shandong Province were taken as examples to study under different emission reduction options and risk-aversion levels in Section 4. The model proposed could be widely applicable to other regions in China for energy system optimization and planning, and the results obtained could provide decision schemes for energy system planning and pollutant mitigation in the mid/long-term period.

## 2. Energy System Analysis of Shandong Province

Shandong Province, which is located in China's east coast ($34°22.9'~38°24.01'$ N, $114°47.5'~122°42.3'$ E), is one of the most crucial energy consumption and production regions (as shown in Figure 1). Shandong Province covers an area of 157,100 km$^2$ with 17 districts. The resident population was 100.5 million in 2018. Most parts of Shandong are located in the warm temperate zone. The GDP of Shandong Province increased by 6.4%, corresponding to CNY 7.6 trillion, which contributed to the third largest economy area in China in 2018.

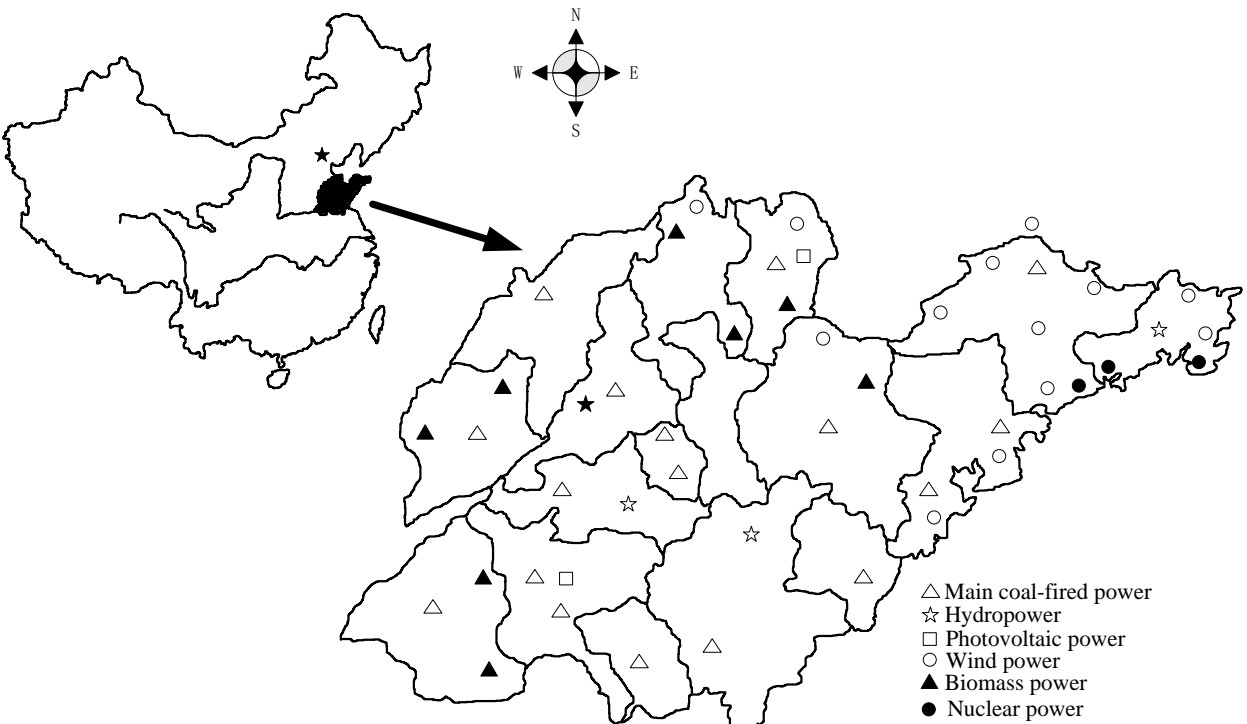

**Figure 1.** Geographical location of Shandong Province.

Due to the intensified variation of energy activities caused by the economic development mode, new-type urbanization, and industrial structure adjustment, the statistics of energy consumption has displayed a rapid growth trend in Shandong Province. According to the Shandong Province Statistical Yearbook (2010–2018), the terminal energy consumption increased from 302.4 million ton of coal equivalent in 2010 to 386.8 million ton of coal equivalent in 2017, with the consumption rates of 70.5%, 17.0%, and 1.9% for coal, crude oil, and primary electricity, respectively. Moreover, it is estimated that the consumption of coal reached to 272.6 million ton in 2017 and the production of coal was only 97.9 million ton, accounting for 35.9% of the total coal consumption, which may lead to serious issues regarding energy-supply security and economic development. Similarly, the consumption of electricity had increased from $329.9 \times 10^3$ GWh in 2010 to $543.0 \times 10^3$ GWh in 2017. By the end of 2018, the total electricity generation capacities had reached 131.1 GW; the proportion of the thermal power installed capacity was 79.10%; and the capacity of new energy power (e.g., wind power, nuclear power, biomass power, and photovoltaic power) and hydropower accounted for 23.2% and 0.8% of the total generation capacities, respectively. These phenomena could reveal that although the energy end-consumption

structure has been improved greatly in recent years, energy consumption still heavily relies on coal in Shandong Province. Obviously, faced with the crisis of energy shortage, the development and utilization of renewable energy resources has been inevitable.

Furthermore, as one of the most challenging issues of the electric power system, air pollution mitigation brings profound impacts on energy-environmental contradictions [9]. Electric power plants have become one of the main sources of air pollution in Shandong Province because of their impact on coal consumption. For instance, in 2017, the amount of $SO_2$, $NO_x$, and PM emission were $739.1 \times 10^3$ ton, $1158.6 \times 10^3$ ton, and $549.6 \times 10^3$ ton, respectively. According to the Shandong Province Social Development Thirteenth Five-Year Plan, large-scale renewable energy projects will be carried out. A total of 30.1 GW of new and renewable energy source (e.g., nuclear power, solar power, biomass power, and wind power) generation facilities will be installed by the end of 2020. In terms of the long-term and mid-term development plans for pollutant and carbon emission control, the province's $SO_2$ emission in 2020 decreased by 127.0% compared to that of 2015 and $NO_x$ emissions were controlled below $1040.0 \times 10^3$ ton in 2020.

In view of the above, it can be found that Shandong Province still faces significant challenges in terms of ensuring the safety of the regional power supply, achieving the goals of air pollution (i.e., PM, $SO_2$ and $NO_x$) mitigation, and optimizing the electric power structure because of the single-power energy structure. These problems have severely hampered the sustainable development of electric power systems for Shandong Province in the future. Generally, electric power systems are complicated with various interrelated electric-related interactions including related to its production, import/export, expansion, consumption, and pollution reduction. In a long-term planning period perspective, the future electricity demand is often modeled as an uncertain parameter with a probability distribution and many key components of the electric power management system contain uncertainties in terms of, for example, energy source availabilities, electricity demands, processing costs, and different power generation technologies. In addition, air pollution emissions (e.g., $SO_2$, $NO_x$, and PM) incorporated within electric power planning systems should be controlled for below a certain tolerance limitation through the adoption of various proper inequality constraints. Therefore, it is essential to study the uncertainties and complexities of electric power system for Shandong Province in the future in order to resolve the contradiction between the optimization of electric power systems and the mitigation of air pollutant emissions.

## 3. Energy System Optimization Model

### 3.1. Optimization Method

Figure 2 presents the general framework of the optimization method, which is integrated with multi-stage stochastic programming, interval-parameter programming, and CVaR techniques. Among these lay uncertain information that is presented as interval numbers, which can be reflected through interval-parameter programming [25], while random information system dynamic characters and policy implications can be effectively addressed by multi-stage stochastic programming [26]. In addition, the system risk introduced by the model uncertainty and the influence of random disturbance on the system can be addressed with the CVaR method [27–29]. The optimization method can be expressed as

$$Min \ f^{\pm} = \sum_{t=1}^{T} \sum_{j=1}^{n} c_{jt}^{\pm} x_{jt}^{\pm} + \sum_{t=1}^{T} \sum_{j=1}^{n} \sum_{h=1}^{h_t} p_{jth} d_{jt}^{\pm} y_{jth}^{\pm} + \lambda \left\{ \sum_{t=1}^{t} \left( \xi_t^{\pm} + \frac{1}{1-\alpha} \sum_{h=1}^{h_t} p_{th}^{\pm} v_{th}^{\pm} \right) \right\} \quad (1a)$$

which is subject to

$$\sum_{j=1}^{n} a_{rjt}^{\pm} x_{jt}^{\pm} \leq b_{rt}^{\pm}, \forall r = 1, 2, \dots, m_1; t = 1, 2, \dots, T \quad (1b)$$

$$\sum_{j=1}^{n} a_{jt}^{\pm} x_{jt}^{\pm} - \sum_{j=1}^{n} a'_{jt} y_{jth}^{\pm} \geq \omega_{th}^{\pm}, \forall h = 1, 2, \ldots, h_t; \ t = 1, 2, \ldots, T \tag{1c}$$

$$v_{th}^{\pm} \geq \sum_{j=1}^{n} c_{jt}^{\pm} x_{jt}^{\pm} + \sum_{j=1}^{n} d_{jt}^{\pm} y_{jth}^{\pm} - \zeta_t^{\pm}, \forall h = 1, 2, \ldots, h_t; t = 1, 2, \ldots, T \tag{1d}$$

$$v_{th}^{\pm} \geq 0, \forall h = 1, 2, \ldots, h_t; t = 1, 2, \ldots, T \tag{1e}$$

$$x_{jt}^{\pm} \geq 0, \forall \ j = 1, 2, \ldots, n; t = 1, 2, \ldots, T \tag{1f}$$

$$y_{jth}^{\pm} \geq 0, \forall j = 1, 2, \ldots, n; \ h = 1, 2, \ldots, h_t; t = 1, 2, \ldots, T \tag{1g}$$

where $c_{jt}^{\pm}$ and $d_{jt}^{\pm}$ are the interval parameters in the objective function; $a_{rjt}^{\pm}, b_{rt}^{\pm}, a_{jt}^{\pm}, a'_{jt}$, and $\omega_{th}^{\pm}$ denote a set of interval parameters in the constraints; $x_{jt}^{\pm}$ are the first stage decision variables; and $y_{jth}^{\pm}$ represent the variables in the second stage. $v_{th}^{\pm}$ are the random variables in the constraints; $p_{jth}$ is the parameter of the probability levels in the second stage; and $\sum P_{jth} = 1$. According to the role of these different parameters in the proposed method, the optimization model, coupled with the RIMSP method, can be established to solve the optimization problem.

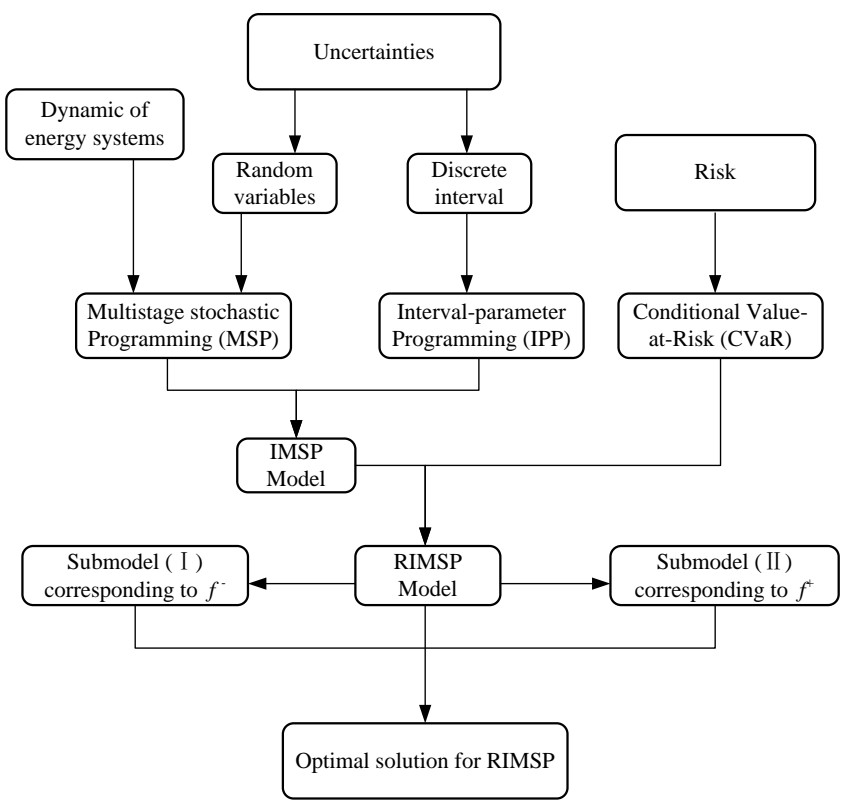

**Figure 2.** General framework of the RIMSP method.

*3.2. Model Development*

The purpose of this paper is to develop a short-term hybrid energy system robust optimization model for regional energy system planning and air pollution mitigation through a case study in Shandong. Six power conversion technologies, six power demand sectors, and three pollutants were considered based on the regional energy system feature and energy development plans. In detail, the model size and boundary conditions were designed as follows: (1) energy resources included coal, hydro-energy, wind, solar, biomass, and nuclear power that correspond to coal-fired power, hydropower, wind power, photovoltaic power, and biomass power (nuclear power was considered as the main electric power

generation technology in the model); (2) power load demand from multiple end-users (e.g., agricultural, industrial, transportation, commercial, and residential sectors) was taken into consideration and divided into three levels (e.g., low, medium, and high) with the corresponding probabilities; (3) PM, SO$_2$, and NOx are the major air pollutants in energy resource conversation processes and the total emission amount control was carried out to improve the environmental quality; and (4) the planning horizon is 6 years and was further divided into two planning periods. The objective of this section was to minimize the system cost (i.e., energy resources cost, power generation cost, import power cost, and air pollutants treatment cost) cover over the whole planning period. The constraints were used to maintain a balance between the supply/demand, resources/final-product, cost/risk, and within the interaction in environment–economic energy system development. Moreover, the inexact multi-stage stochastic integer programming and conditional value-at-risk method were selected to effectively address uncertainties and complexities in the energy system optimization and planning. Based on the roles and definitions of different parameters in the proposed RIMSP method (Section 3.1), the energy system optimization model can be expressed as follows:

$$Minf^{\pm} = f_1^{\pm} + f_2^{\pm} + f_3^{\pm} + f_4^{\pm} + f_5^{\pm} \tag{2a}$$

(1)　Cost for resource consumption:

$$f_1^{\pm} = \sum_{k=1}^{6} \sum_{t=1}^{2} Z_{kt}^{\pm} \cdot PEC_{kt}^{\pm} \tag{2b}$$

(2)　Cost for electricity generation:

$$
\begin{aligned}
f_2^{\pm} = & \sum_{k=1}^{6} \sum_{t=1}^{2} PV_{kt}^{\pm} \cdot XE_{kt}^{\pm} + \sum_{k=1}^{6} \sum_{t=1}^{2} \sum_{h=1}^{H_t} p_{th} \cdot \left( PV_{kt}^{\pm} + PP_{kt}^{\pm} \right) \cdot EQ_{kth}^{\pm} \\
& + \sum_{k=1}^{6} \sum_{t=1}^{2} \sum_{h=1}^{H_t} p_{th} \cdot \left( YCA_{kth}^{\pm} \cdot A_{kt}^{\pm} + XCA_{kth}^{\pm} \cdot B_{kt}^{\pm} \right)
\end{aligned}
\tag{2c}
$$

(3)　Net import cost for electric power:

$$f_3^{\pm} = \sum_{t=1}^{2} \left( IE_t^{\pm} \cdot IPE_t^{\pm} - EE_t^{\pm} \cdot EPE_t^{\pm} \right) \tag{2d}$$

(4)　Cost for air pollutant emission:

$$
\begin{aligned}
f_4^{\pm} = & \sum_{s=1}^{3} \sum_{k=1}^{6} \sum_{t=1}^{2} XE_{kt}^{\pm} \cdot \xi_{skt}^{\pm} \cdot \left( 1 - \eta_{skt}^{\pm} \right) \cdot CPC_{st}^{\pm} \\
& + \sum_{s=1}^{3} \sum_{k=1}^{6} \sum_{t=1}^{2} \sum_{h=1}^{H_t} p_{th} \cdot EQ_{kth}^{\pm} \cdot \xi_{skt}^{\pm} \cdot \left( 1 - \eta_{skt}^{\pm} \right) \cdot DPC_{st}^{\pm}
\end{aligned}
\tag{2e}
$$

(5)　Risk-aversion of the electric power system:

$$f_5^{\pm} = \lambda \cdot \sum_{t=1}^{2} \left( VaR_t^{\pm} + \frac{1}{1-\alpha} \sum_{h=1}^{H_t} p_{th} \cdot \varsigma_{th}^{\pm} \right) \tag{2f}$$

where $k$ denotes electric power generation technology; $k$ = 1, 2, 3, 4, 5, and 6 for coal-fired power, hydropower, solar power, wind power, biomass power, and nuclear power; $t$ denotes the planning period; $t$ = 1 represents period 1 from 2019 to 2021 and $t$ = 2 represents period 2 from 2022 to 2024; $s$ represents the atmospheric pollutants, wherein $s$ = 1, 2, and 3 represent SO$_2$, NOx, and PM; $h$ is electricity-demand level; and lastly $i$ represents the power load demand sectors, wherein $i$ = 1, 2, 3, 4, 5, and 6 represent the agriculture, industry, building industry, transportation, business, and residential department.

These variables are subject to:

(1) constraints regarding the electricity supply and demand balance, expressed as

$$\sum_{k=1}^{6} \left( XE_{kt}^{\pm} + EQ_{kth}^{\pm} \right) + IE_{t}^{\pm} - EE_{t}^{\pm} \geq \sum_{i=1}^{6} AD_{ith}^{\pm}, \forall t, h; \tag{2g}$$

$$IE_{t}^{\pm} \leq \gamma_{t} \cdot \sum_{i=1}^{6} AD_{ith}^{\pm}, \forall t, h; \tag{2h}$$

(2) constraints regarding the environment capacity, expressed as

$$\sum_{k=1}^{6} \left( XE_{kt}^{\pm} + EQ_{kth}^{\pm} \right) \cdot \zeta_{kst}^{\pm} \cdot \left( 1 - \eta_{kst}^{\pm} \right) \leq MAGE_{st}^{\pm}, \forall s, t, h; \tag{2i}$$

(3) constraints regarding the electric power production capacity, expressed as

$$XE_{kt}^{\pm} + EQ_{kth}^{\pm} \leq ST_{kt}^{\pm} \cdot \left\{ ICP_{k}^{\pm} + \sum_{t=1}^{T} XCA_{kth}^{\pm} \right\}, \forall k, t, h; \tag{2j}$$

$$XCA_{kth}^{\pm} \leq M_{kt}^{\pm} \cdot YCA_{kth}^{\pm}, \forall k, t, h; \tag{2k}$$

$$XE_{kt}^{\pm} \geq EQ_{kth}^{\pm}, \forall k, t, h; \tag{2l}$$

(4) constraints regarding the capacity expansion, expressed as

$$YCA_{kth}^{\pm} \begin{cases} = 1, \text{if capacity expansion is undertaken} \\ = 0, \text{if otherwise} \end{cases}, \forall k, t, h; \tag{2m}$$

(5) constraints regarding the coal mass balance, expressed as

$$\left( XE_{1t}^{\pm} + EQ_{1th}^{\pm} \right) \cdot FE_{1t}^{\pm} \leq Z_{1t}^{\pm}, \forall t, h; \tag{2n}$$

(6) constraints regarding the hydropower mass balance, expressed as

$$\left( XE_{2t}^{\pm} + EQ_{2th}^{\pm} \right) \cdot FE_{2t}^{\pm} \leq Z_{2t}^{\pm}, \forall t, h; \tag{2o}$$

(7) constraints regarding the solar mass balance, expressed as

$$\left( XE_{3t}^{\pm} + EQ_{3th}^{\pm} \right) \cdot FE_{3t}^{\pm} \leq Z_{3t}^{\pm}, \forall t, h; \tag{2p}$$

(8) constraints regarding the wind mass balance, expressed as

$$\left( XE_{4t}^{\pm} + EQ_{4th}^{\pm} \right) \cdot FE_{4t}^{\pm} \leq Z_{4t}^{\pm}, \forall t, h; \tag{2q}$$

(9) constraints regarding the biomass mass balance, expressed as

$$\left( XE_{5t}^{\pm} + EQ_{5th}^{\pm} \right) \cdot FE_{5t}^{\pm} \leq Z_{5t}^{\pm}, \forall t, h; \tag{2r}$$

(10) constraints regarding the nuclear mass balance, expressed as

$$\left( XE_{6t}^{\pm} + EQ_{6th}^{\pm} \right) \cdot FE_{6t}^{\pm} \leq Z_{6t}^{\pm}, \forall t, h; \text{and} \tag{2s}$$

(10) constraints regarding risk control, expressed as

$$
\begin{aligned}
&\sum_{k=1}^{6} Z_{kt}^{\pm} \cdot PEC_{kt}^{\pm} + \sum_{k=1}^{6} PV_{kt}^{\pm} \cdot XE_{kt}^{\pm} + \sum_{k=1}^{6} \left(PV_{kt}^{\pm} + PP_{kt}^{\pm}\right) \cdot EQ_{kth}^{\pm} \\
&+ \sum_{k=1}^{6} \left(YCA_{kth}^{\pm} \cdot A_{kt}^{\pm} + XCA_{kth}^{\pm} \cdot B_{kt}^{\pm}\right) + \left(IE_{t}^{\pm} \cdot IPE_{t}^{\pm} - EE_{t}^{\pm} \cdot EPE_{t}^{\pm}\right) \\
&+ \sum_{s=1}^{3} \sum_{k=1}^{6} XE_{kt}^{\pm} \cdot \xi_{skt}^{\pm} \cdot \left(1 - \eta_{skt}^{\pm}\right) \cdot CPC_{st}^{\pm} + \sum_{s=1}^{3} \sum_{k=1}^{6} EQ_{kth}^{\pm} \cdot \xi_{skt}^{\pm} \cdot \left(1 - \eta_{skt}^{\pm}\right) \cdot DPC_{st}^{\pm} \\
&- VaR_{t}^{\pm} \leq \varsigma_{th}^{\pm}
\end{aligned}
\tag{2t}
$$

Decision variable

$Z_{kt}^{\pm}$ is the energy resource consumption amount for electricity generation (PJ); $XE_{kt}^{\pm}$ is fixed power generation target ($10^3$ GWh); $XCA_{kth}^{\pm}$ denotes the expanded capacity (GW); $EQ_{kth}^{\pm}$ represents the excess power generation amount ($10^3$ GWh); $YCA_{kth}^{\pm}$ denotes the capacity-expansion option; and $IE_{t}^{\pm}$ and $EE_{t}^{\pm}$ are the imported and exported electricity amount ($10^3$ GWh).

Parameters

$PEC_{kt}^{\pm}$ is purchase cost for the energy resource (USD million/PJ); $PV_{kt}^{\pm}$ denotes the regular operation cost for power generation (USD million/$10^3$ GWh); $PP_{pt}^{\pm}$ denotes the surplus cost for power generation (USD million/$10^3$ GWh); $A_{pt}^{\pm}$ is the fixed-charge for the power conversion expansion (USD million); $B_{pt}^{\pm}$ denotes the variable cost for the power conversion expansion (USD million/GW); $IPE_{t}^{\pm}$ and $EPE_{t}^{\pm}$ are the price of imported and exported electricity (USD million/$10^3$ GWh); $CPC_{st}^{\pm}$ is the cost for pollutant treatment (USD million/$10^3$ tonne); $\xi_{kst}^{\pm}$ is the emission coefficient of pollutants from conversion technology ($10^3$ tonne/$10^3$ GWh); $DPC_{it}^{\pm}$ denotes the punishment cost for pollutant treatment (USD million/$10^3$ tonne); $AD_{dth}^{\pm}$ is the power load demand of the sector (GWh); $\gamma$ is the proportion of imported electricity; $\eta_{ipt}^{\pm}$ denotes removal coefficient of pollutant $i$ in period $t$; $\xi_{ipt}^{\pm}$ is emission coefficient of pollutant $i$ from conversion technology $p$ in period $t$ (kilotonnes/GWh); $MAGE_{it}^{\pm}$ represents the allowable upper bounds of the pollutant emission quantity in period $t$ (kilotonnes); $M_{pt}^{\pm}$ is upper bound of the expansion capacity for conversion technology $p$ in period $t$ (GW); $ST_{pt}^{\pm}$ denotes the service time of conversion technology $p$ in period $t$ (h); $ICP_{pt}$ is the initial capacity for conversion technology $p$ in period $t$ (GW); $Z_{1t}^{\pm}$ represents the domestic coal supply in period $t$ (PJ); $Z_{2t}^{\pm}$ denotes the electricity production by hydropower in period $t$ (PJ); $Z_{3t}^{\pm}$ is the domestic solar power supply in period $t$ (PJ); $Z_{4t}^{\pm}$ denotes the domestic wind power supply in period $t$ (PJ); $Z_{5t}^{\pm}$ represents the domestic biomass supply in period $t$ (PJ); $Z_{6t}^{\pm}$ denotes the domestic nuclear supply in period $t$ (PJ); and lastly $FE_{pt}^{\pm}$ denotes the energy consumption of conversion technology $p$ in period $t$ (TJ/GWh).

### 3.3. Data Collection and Scenarios Definition

The study system covers a time horizon of 6 years (2019–2024), which is divided into two planning periods with each representing a 3-year span (i.e., period 1: 2019, 2020, and 2021; period 2: 2022, 2023, and 2024). From the previous analysis, the existing power generation system in Shandong Province cannot meet electricity demands and needs to purchase electricity from other places or expand the generation capacities. Through analyzing a series of studies for the regional power system, many economic-power data were acquired, such as from the Shandong Statistics Bureau (2010–2018) [30], Shandong Thirteenth Five-Year Plan, and Shandong Electric Thirteenth Five-Year Plan. Table 1 provides the economic and technological dates of different conversion technologies in detail, which include the regular and surplus costs for power generation, the fixed and variable costs for capacity expansion, and the operation times of each conversion technology.

**Table 1.** Economic and technological data for each conversion technology.

| Conversion Technology | | Time Period | |
|---|---|---|---|
| | | $t = 1$ | $t = 2$ |
| Regular and surplus costs for power generation of each power conversion technology (USD $10^3$/GWh) | | | |
| Coal-fired power | Regular cost, $PV_{1t}^{\pm}$ | [23.1, 25.4] | [23.9, 26.2] |
| | Surplus cost, $PP_{1t}^{\pm}$ | [9.9, 10.5] | [10.4, 11.1] |
| Hydropower | Regular cost, $PV_{2t}^{\pm}$ | [13.1,14.6] | [12.4, 13.9] |
| | Surplus cost, $PP_{2t}^{\pm}$ | [5.6, 6.6] | [5.8, 7.2] |
| Photovoltaic power | Regular cost, $PV_{3t}^{\pm}$ | [8.5, 9.4] | [8.3, 9.0] |
| | Surplus cost, $PP_{3t\pm}$ | [3.6, 4.0] | [3.6, 4.4] |
| Wind power | Regular cost, $PV_{4t}^{\pm}$ | [6.4, 8.6] | [6.2, 8.2] |
| | Surplus cost, $PP_{4t}^{\pm}$ | [2.9, 3.2] | [3.1, 3.4] |
| Biomass power | Regular cost, $PV_{5t}^{\pm}$ | [28.3, 30.0] | [29.9, 31.7] |
| | Surplus cost, $PP_{5t}^{\pm}$ | [14.7, 15.6] | [15.9, 16.2] |
| Nuclear power | Regular cost, $PV_{6t}^{\pm}$ | [7.2, 8.5] | [6.8, 8.1] |
| | Surplus cost, $PP_{6t}^{\pm}$ | [3.5, 4.0] | [3.2, 3.7] |
| Fixed (USD $10^6$) and variable (USD $10^6$/GW) costs for capacity expansion | | | |
| Coal-fired power | Fixed cost, $A_{1t}^{\pm}$ | [816.0, 96.0] | [757.4, 891.0] |
| | Variable cost, $B_{1t}^{\pm}$ | [544.2, 640.3] | [522.5, 614.9] |
| Hydropower | Fixed cost, $A_{2t}^{\pm}$ | [2065.5, 2430.0] | [2021.3, 2378.0] |
| | Variable cost, $B_{2t}^{\pm}$ | [1380.6, 1624.2] | [1394.4, 1640.4] |
| Photovoltaic power | Fixed cost, $A_{3t}^{\pm}$ | [2159.0, 2540.0] | [1981.4, 2331.0] |
| | Variable cost, $B_{3t}^{\pm}$ | [1439.5, 1693.6] | [1367.5, 1608.9] |
| Wind power | Fixed cost, $A_{4t}^{\pm}$ | [1704.3, 1583.6] | [2005.0, 1863.0] |
| | Variable cost, $B_{4t}^{\pm}$ | [1137.9, 1338.7] | [1092.4, 1285.2] |
| Biomass power | Fixed cost, $A_{5t}^{\pm}$ | [1856.4, 2184.0] | [1746.8, 2055.0] |
| | Variable cost, $B_{5t}^{\pm}$ | [1210.0, 1423.5] | [1178.7, 1386.7] |
| Nuclear power | Fixed cost, $A_{6t}^{\pm}$ | [2082.5, 2450.0] | [1787.6, 2103.0] |
| | Variable cost, $B_{6t}^{\pm}$ | [1389.8, 1635.0] | [1232.5, 1450.0] |
| Operation time for generation technology $p$ in period $t$ (h) | | | |
| Coal-fired power | $ST_{1t}^{\pm}$ | [16,800, 17,400] | [16,800, 17400] |
| Hydropower | $ST_{2t}^{\pm}$ | [11,700, 12,000] | [11,700, 12,000] |
| Photovoltaic power | $ST_{3t}^{\pm}$ | [2700, 3000] | [2700, 3000] |
| Wind power | $ST_{4t}^{\pm}$ | [8700, 9000] | [8700, 9000] |
| Biomass power | $ST_{5t}^{\pm}$ | [13,800, 14,400] | [13,800, 14,400] |
| Nuclear power | $ST_{6t}^{\pm}$ | [18,900, 19,200] | [18,900, 19,200] |

According to the Shandong Province Statistical Yearbook (2010–2018), three discrete target values of power load demand (i.e., low, medium, and high) were selected. As shown in Table 2, to illustrate the applicability of the developed model, electricity demands of different end users were assumed to be uncertain with three probability levels (i.e., 20% for the low level of electricity demand, 60% for the medium level of electricity demand, and 20% for the high level of electricity demand, respectively). According to the medium and long-term planning of energy development and environmental-emission reduction of Shandong Province, three levels of emission reduction targets were considered, corresponding to reductions of 0%, 7%, and 15% of the total air pollution emissions during the planning period. In this study, in order to have decision-makers determine the value of the risk aversion and to find points from the frontier, we built an approximation of the efficient frontier by setting three $\lambda$ values (i.e., 0.05, 0.5, and 10, respectively). Thus, to compare the effects of different risk aversion parameter $\lambda$ and the emission reduction targets on the

regional electricity supply strategies, nine different scenarios were designed (as shown in Table 3).

**Table 2.** End user's total electricity demand.

| Demand Sector | Demand Level | Probability (%) | Electricity Demand ($10^3$ GWh) | |
| --- | --- | --- | --- | --- |
| | | | *t* = 1 | *t* = 2 |
| Agriculture | L | 20 | [25.4, 25.9] | [31.7, 32.8] |
| | M | 60 | [27.9, 28.4] | [34.7, 35.9] |
| | H | 20 | [30.3, 30.8] | [37.7, 39.0] |
| Industrial | L | 20 | [864.0, 880.5] | [1075.6, 1113.8] |
| | M | 60 | [946.6, 963.2] | [1178.5, 1218.4] |
| | H | 20 | [1029.3, 1045.8] | [1281.5, 1323.0] |
| Building industry | L | 20 | [9.8, 10.0] | [12.2, 12.7] |
| | M | 60 | [10.8, 10.9] | [13.38, 13.8] |
| | H | 20 | [11.7, 11.9] | [14.6, 15.0] |
| Transportation | L | 20 | [17.6, 17.9] | [21.9, 22.7] |
| | M | 60 | [19.3, 19.5] | [24.0, 24.8] |
| | H | 20 | [21.0, 21.3] | [26.1, 27.0] |
| Business | L | 20 | [82.1, 83.7] | [102.2, 105.8] |
| | M | 60 | [90.0, 91.5] | [112.0, 115.8] |
| | H | 20 | [97.8, 99.4] | [121.8, 125.7] |
| Residential | L | 20 | [122.2, 124.5] | [152.1, 157.5] |
| | M | 60 | [133.9, 136.2] | [166.7, 172.3] |
| | H | 20 | [145.6, 147.9] | [181.3, 187.1] |

Note: the symbols L, M, and H denote that the total electricity demand is low, medium, and high in period 1, respectively.

**Table 3.** Nine analysis scenarios.

| Scenario | Risk Aversion Parameter | Emission Reduction Target |
| --- | --- | --- |
| Scenario 1 (S_1) | 0.05 | 0% |
| Scenario 2 (S_2) | 0.05 | 7% |
| Scenario 3 (S_3) | 0.05 | 15% |
| Scenario 4 (S_4) | 0.5 | 0% |
| Scenario 5 (S_5) | 0.5 | 7% |
| Scenario 6 (S_6) | 0.5 | 15% |
| Scenario 7 (S_7) | 10 | 0% |
| Scenario 8 (S_8) | 10 | 7% |
| Scenario 9 (S_9) | 10 | 15% |

## 4. Results and Discussion

The objective function of the interval multi-stage stochastic integer programming model is to minimize the expected cost and to establish a stable budget under different scenarios of both pollutant reduction targets and risk-aversion levels over the planning horizon. Interval solutions can provide energy utilization schemes and help managers acquire multiple decision alternatives, which are useful for decision-makers to obtain insight regarding tradeoffs between environmental and economic objectives.

### 4.1. Optimized Electricity Generation in Different Emission Reduction Scenarios

From the above results in Section 2, Shandong Province experiences serious air pollution generation from electric power systems. In this study, three emission reduction levels of air pollution (i.e., $SO_2$, $NO_x$, and PM) were taken into consideration, including 0%, 7%, and 15%. Figures 3 and 4 present the results of the optimized electricity generation plans for different power conversion technologies under different emission reduction levels over the planning horizon. Generally, the proportion of the amount of coal-fired power generation would be reduced, with significant emission reductions [31]. Additionally, lean energy and new energy power generation would be rapidly increased, such as through wind and photovoltaic power generation technologies. For example, in pe-

riod 1, under the medium demand level, the coal-fired power generations calculated to [1738.5, 1771.4] $\times 10^3$ GWh, [1665.0, 1711.6] $\times 10^3$ GWh, and [1673.9, 1715.1] $\times 10^3$ GWh in S_1, S_2, and S_3, respectively. Similarly, the wind power generations calculated to 98.2 $\times 10^3$ GWh, [137.4, 151.1] $\times 10^3$ GWh, and [137.4, 145.6] $\times 10^3$ GWh in S_1, S_2, and S_3, respectively. Furthermore, due to the fact that coal-fired energy accounts for a great amount of pollutant emissions, managers could limit the operation of coal-fired power and encourage both new and clean energy development, especially wind power and nuclear power. In S_2, the hydropower calculated to [41.4, 57.2] $\times 10^3$ GWh, [60.4, 82.8] $\times 10^3$ GWh, and 82.8 $\times 10^3$ GWh when the demand level was low-low, low-medium, and low-high in period 2, respectively. Similarly, wind power calculated to [137.4, 150.5] $\times 10^3$ GWh, [137.4, 150.5] $\times 10^3$ GWh, and 196.3 $\times 10^3$ GWh, while it calculated to [58.5, 65.4] $\times 10^3$ GWh, 65.4 $\times 10^3$ GWh, and 65.4 $\times 10^3$ GWh for nuclear power, respectively.

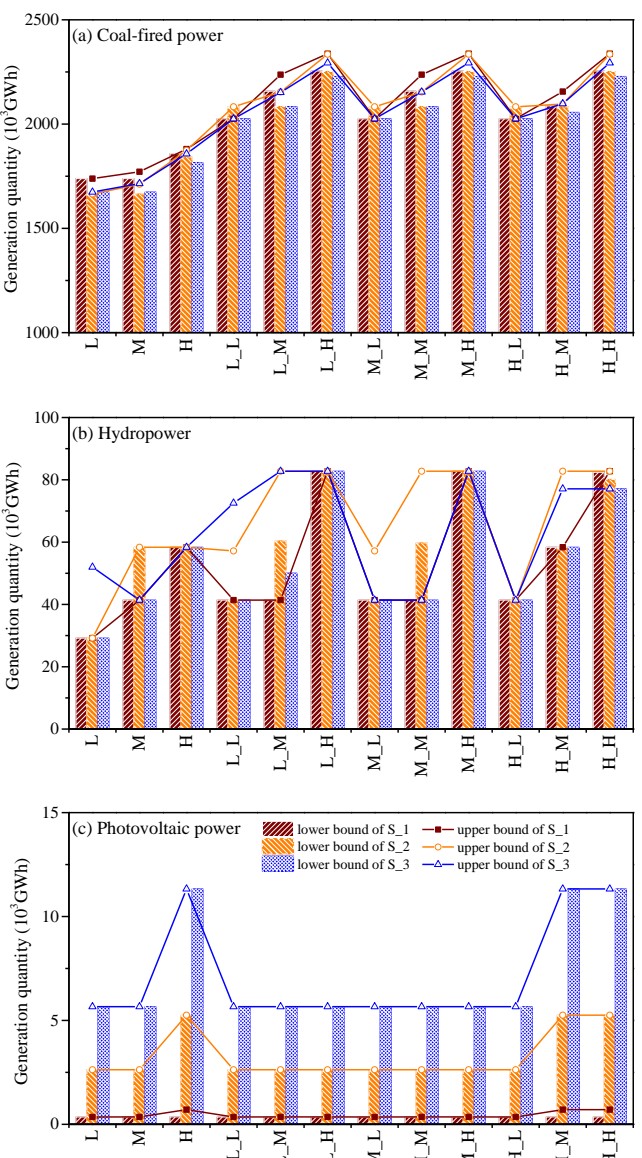

**Figure 3.** Optimized electricity generation for coal-fired power, hydropower, and photovoltaic power in different emission reduction scenarios.

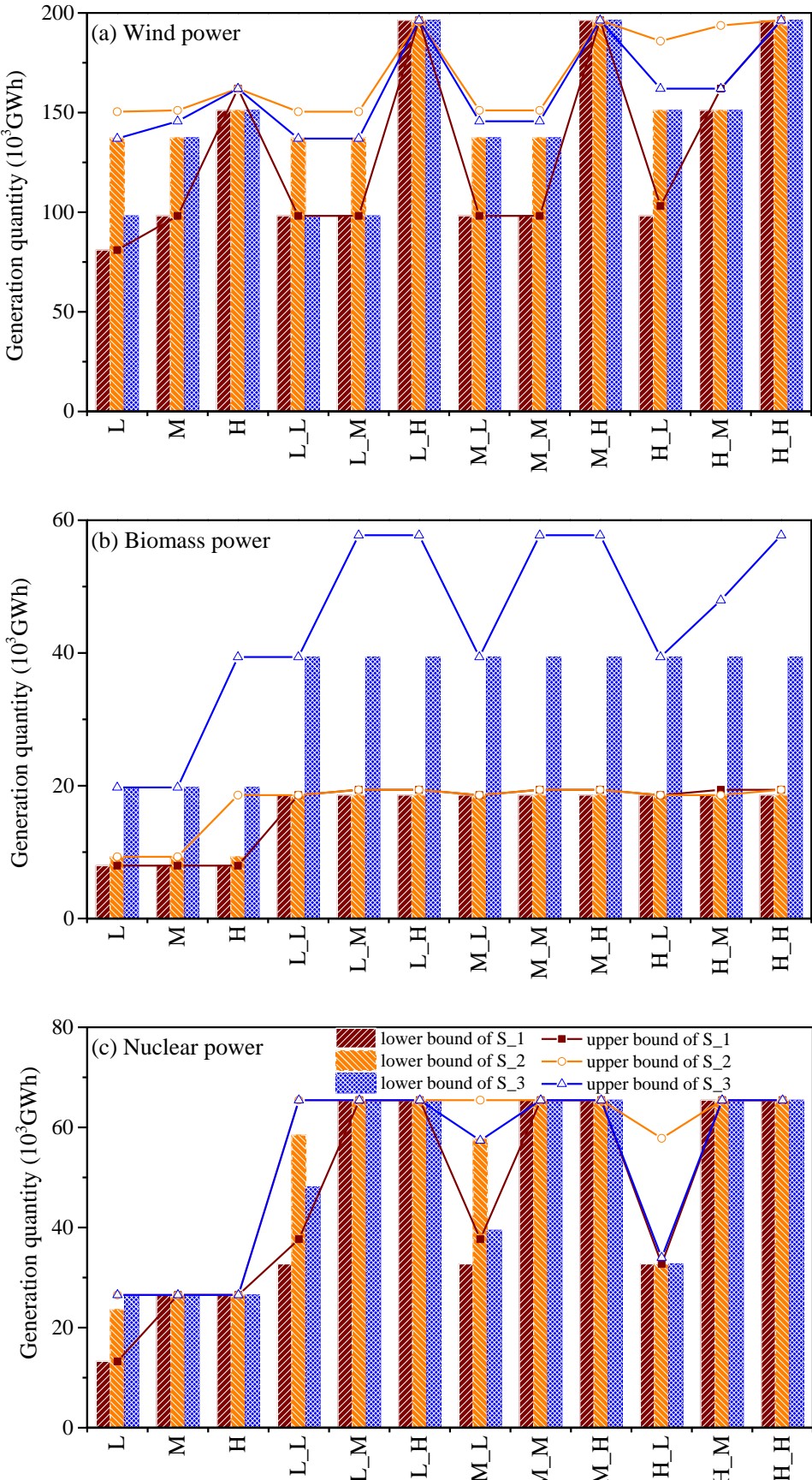

**Figure 4.** Optimized electricity generation for wind power, biomass power, and nuclear power in different emission reduction scenarios.

In addition, the structure of the Shandong power system was relatively unitary, mainly including coal-fired power, while the proportion of clean power was less. In this study, when the pollution-emission reduction rate was more than 15%, the power system could be crashed, and there were no solutions under the condition of minimum economic cost. This phenomenon could be attributed to the fact that the air pollutant mitigation measures for the Shandong power system was modelled by the project emissions, which controlled air pollution emissions by engineering treatment facilities including for desulfurization, denitration, and dust removal [9]. However, with the continuous improvement of the economy, air pollutant emissions would further increase, thus the project emissions will not be enough to totally support the tasks of economic development and pollution-emission reductions [32]. As a result, in order to effectively reduce air pollutant emissions, the development and utilization of new and renewable energy should be strengthened in the future. For example, the Shandong Jiaodong peninsula has good nuclear site resources and three nuclear power plants have been build, including the Haiyang nuclear power plant, Hongshiding nuclear power plant, and Shidaowan nuclear power plant.

### 4.2. Optimized Electricity Generation in Different Risk Aversion Scenarios

Figures 5 and 6 illustrate the results of optimized electricity generation from different power generation technologies in different risk aversion scenarios wherein the emission reduction rate was fixed as 15%. In this study, the CVaR method was integrated in the proposed optimization model as the objective function and constraint to determine a policy that minimizes this measure of risk. The results indicate that electricity generation from various power conversion technologies varies greatly, with risk aversion parameter $\lambda$ fixed as 0.05, 0.5, and 10, respectively. For instance, in period 1, the coal-fired power generations were [1547.5, 1673.9] $\times 10^3$ GWh, [1485.9, 1545.3] $\times 10^3$ GWh, and [1488.0, 1547.5] $\times 10^3$ GWh under the low demand level, while they were calculated as [29.2, 58.4] $\times 10^3$ GWh, [52.0, 58.4] $\times 10^3$ GWh, and [41.4, 58.4] $\times 10^3$ GWh for hydropower, respectively. In general, electricity generations from conventional power conversion technologies would decrease, while those for clean and new power would relatively increase as the risk levels rise. The reasons for these phenomena concern that the conventional coal-fired power conversion technology corresponded to a higher air pollution-emission rate and the clean power conversion technology could reduce the risk of environment pollution [33]. However, in the regional electric power system, coal-fired power accounted for a large proportion of pollutant emissions compared with other power generation technologies [9,10]. From these results, it could be seen that the coal-fired power generations were increased steadily in period 2 and for other power conversion technologies, the power generations would have substantial fluctuations compared to those in the first period. For example, the coal-fired power would be generated with the values of [2228.4, 2293.4] $\times 10^3$ GWh, [2248.9, 2293.6] $\times 10^3$ GWh, and [2224.5, 2293.5] $\times 10^3$ GWh under the low-high demand level in period 2, while for photovoltaic power these were [2.2, 5.7] $\times 10^3$ GWh, [4.3, 5.7] $\times 10^3$ GWh, and [2.2, 4.3] $\times 10^3$ GWh, respectively.

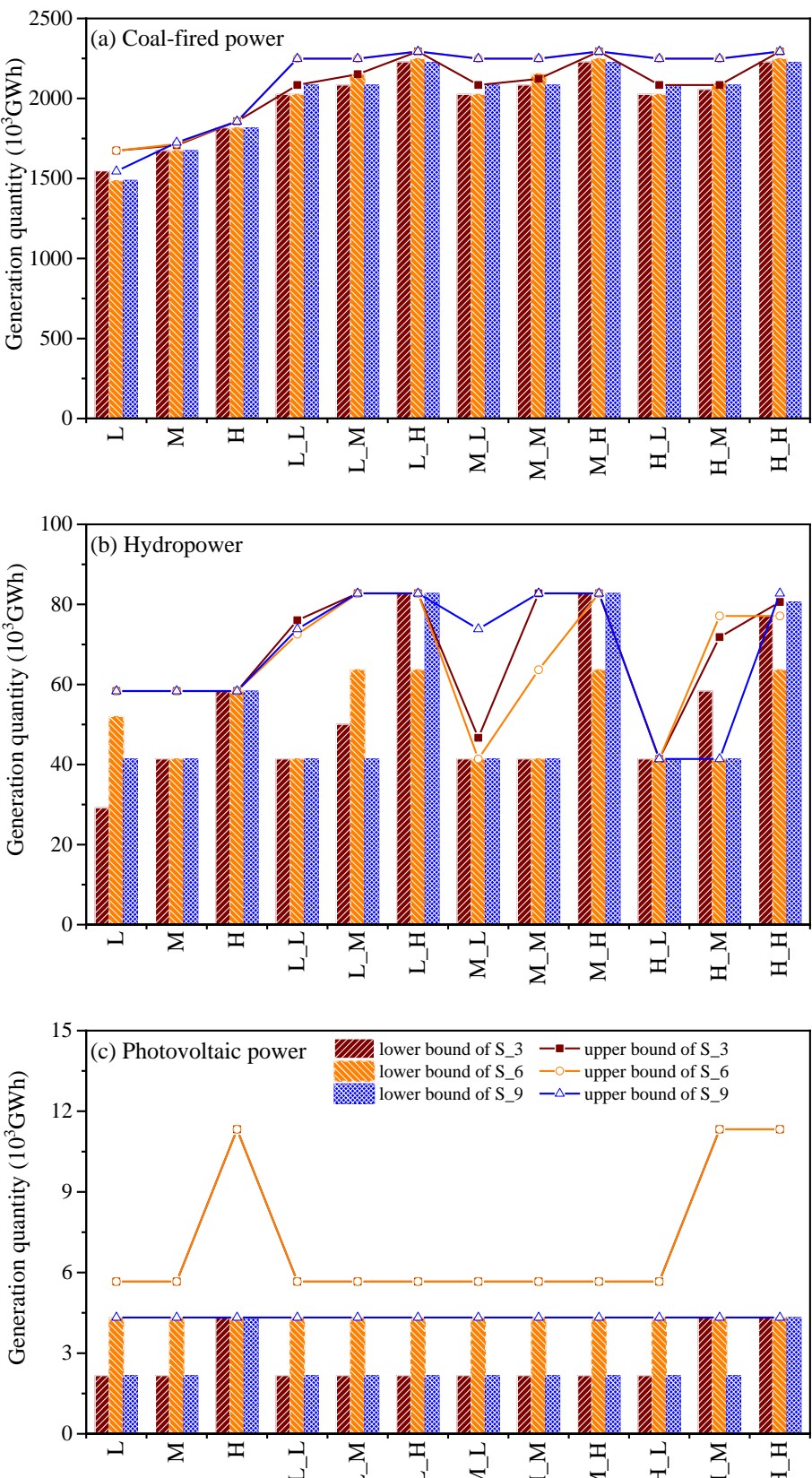

**Figure 5.** Optimized electricity generation for coal-fired power, hydropower, and photovoltaic power in different risk aversion scenarios.

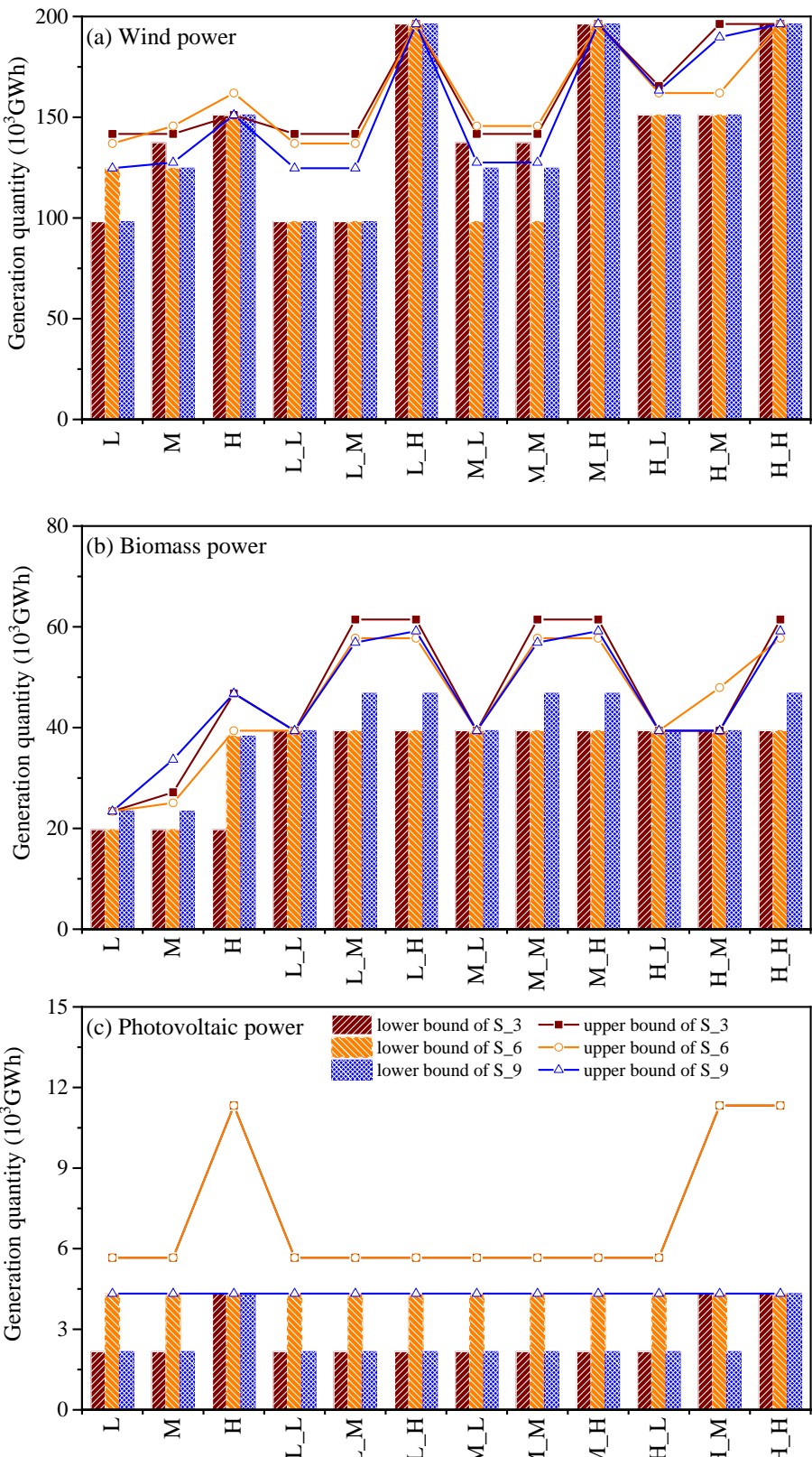

**Figure 6.** Optimized electricity generation for wind power, biomass power, and nuclear power under different risk-aversion levels.

### 4.3. Capacity Expansion Schemes in Different Emission Reduction Scenarios

The capacity expansion schemes under different emission reduction levels were calculated and they are presented in Figures 7 and 8, with the risk aversion parameter $\lambda$ fixed as 0.05. Generally, the existing capacities could not satisfy the future electricity demands from end users, while some capacity expansion projects would be undertaken to avoid the insufficient electricity supply. From the graphs, it can be observed that with the stricter environmental requirements, clean power conversion technologies were key considerations in establishing additional capacities to meet the rising social demands. In detail, in period 1, the expansions were [0.87, 0.92] GW, [0.9, 2.8] GW, and [1.7, 2.8] GW for hydropower, and [2.4, 2.6] GW, [2.4, 2.6] GW, and [6.0, 8.5] GW for wind power under the low, medium, and high demand levels, respectively. Moreover, the nuclear power could be expanded with an increment of [1.0, 1.9] GW, [1.0, 1.9] GW, and [1.9, 2.0] GW in period 1 under the low, medium, and high demand levels, respectively. In addition, the differences included the fact that the photovoltaic power only expanded at a lesser capacity and the biomass power could not be expanded in period 1. The reasons for these phenomena were attributed to the fact that the biomass power could be discharged in large amounts of air pollutants and the photovoltaic power was easily affected by meteorological factors [34,35]. In period 2, there would be eight possible capacity expansion schemes under various electricity demand levels. In comparison, coal-fired power and wind power would be expanded under constraints for controlling air pollutant emissions. Additionally, nuclear power conversion technology had relatively high fixed and variable costs for capacity expansions.

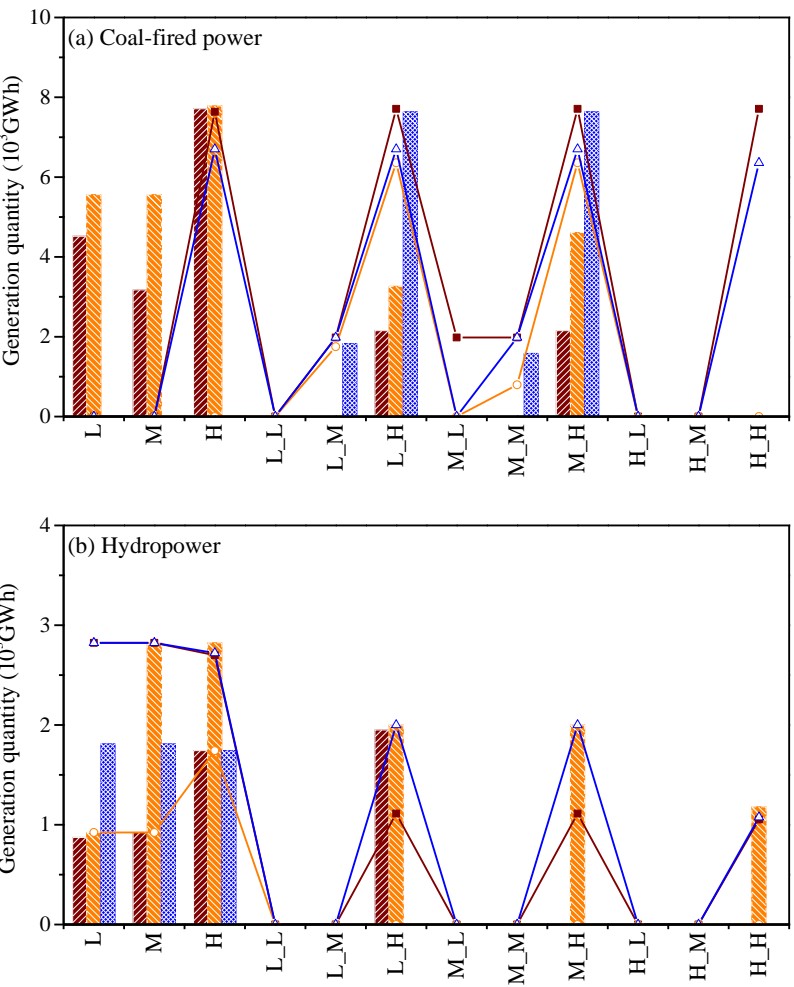

**Figure 7.** *Cont.*

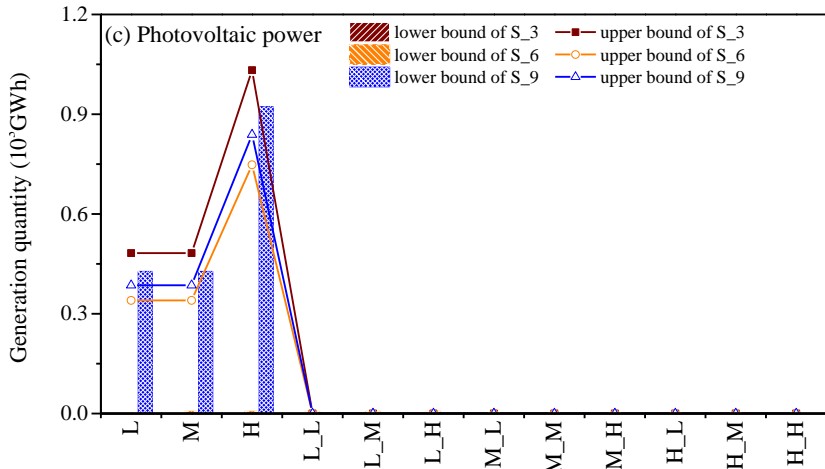

**Figure 7.** Capacity expansion schemes for coal-fired power, hydropower, and photovoltaic power under different emission reduction levels.

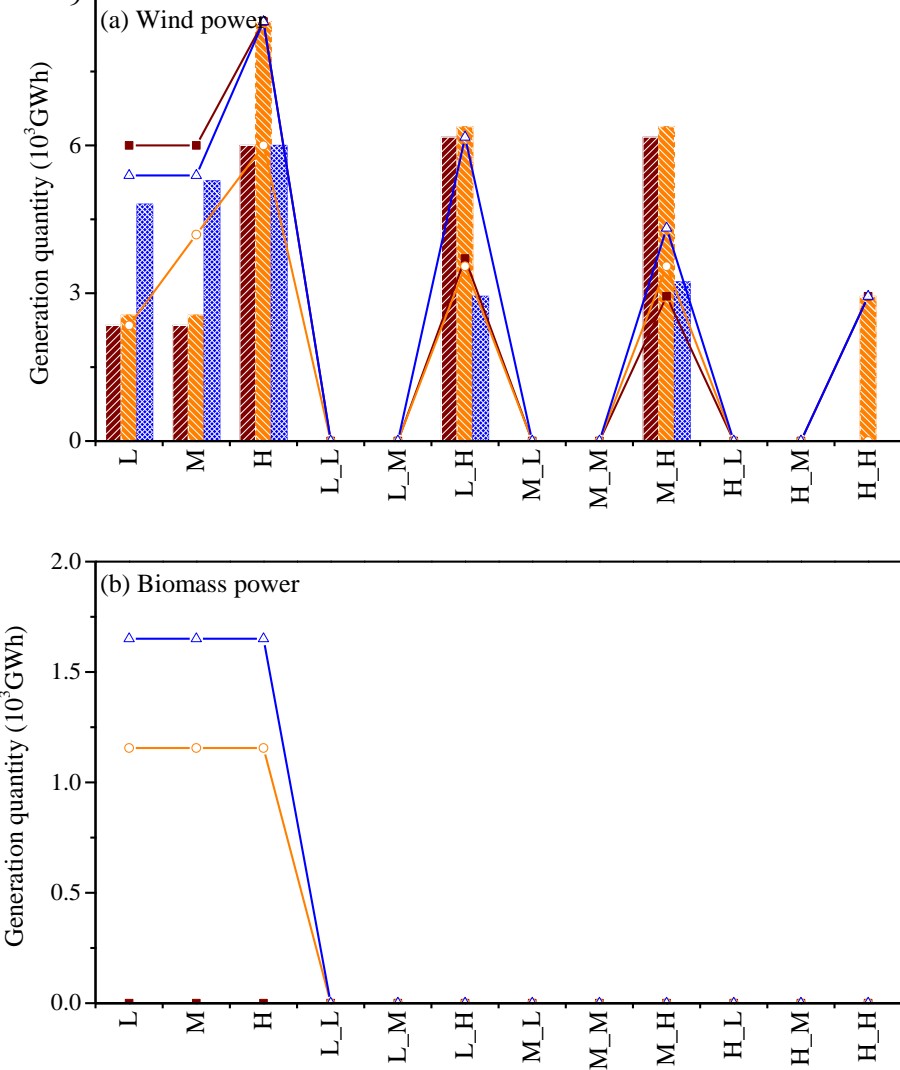

**Figure 8.** *Cont.*

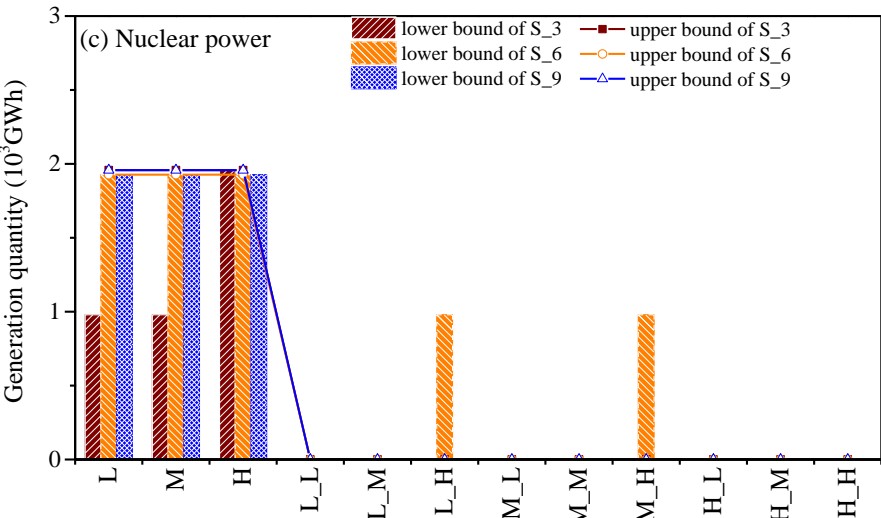

**Figure 8.** Capacity expansion schemes for wind power, biomass power, and nuclear power under different emission reduction levels.

### 4.4. Economic Risk Analysis for the Shandong Province Energy System Model

The objective of the proposed model is to minimize the system cost according to optimized air pollution mitigation and the avoided environment risk in this study. The values of the objective function under the nine scenarios were calculated and are listed in Figure 9. The results indicate that the available amount of pollutant mitigation and the risk-aversion level would not only lead to different power generation schemes but would also have impacts on the total system cost. In detail, planning electric power systems without pollution-emission reduction constraints would lead to a lower system cost; conversely, planning with a high system cost would lead to higher emissions under the fixed risk-aversion level. For example, with the $\lambda$ fixed as 0.05, the total cost would be (in USD) $[828.0, 965.2] \times 10^9$, $[832.5, 974.5] \times 10^9$, and $[472.7, 553.7] \times 10^9$ in the two periods. The costs would have a slight increase in period 2 compared to those in period 1. The main reason for this concerns the increasing electricity demands that could lead to increasing power generations and facility expansions, and there would be more pressures to reduce pollution emissions in period 2. In addition, electric power system management schemes with some lower risk-control constraints would lead to higher system costs, while low system costs would lead to higher risk-control constraints under the fixed emission-reduction rate. For instance, without pollution-emission reduction constraints, the total system cost for scenario 1 (in USD) was $[828.0, 965.2] \times 10^9$, while this calculated to (in USD) $[833.7, 984.5] \times 10^9$ and $[847.2, 985.0] \times 10^9$ for S_4 and S_7, respectively.

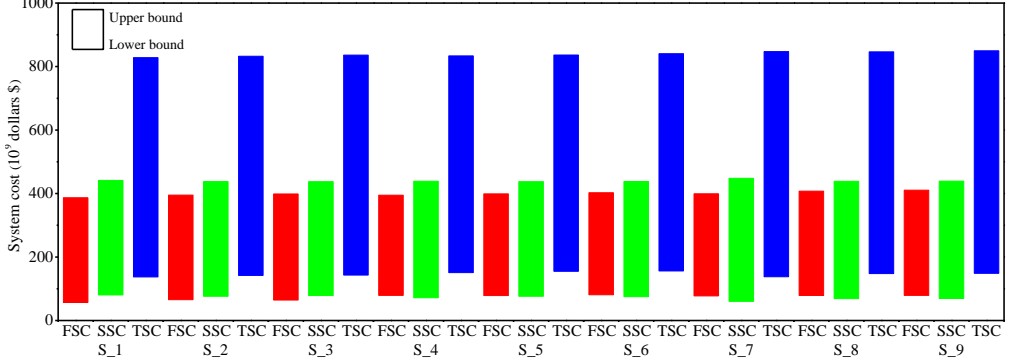

**Figure 9.** System cost under different scenarios. In the plot, the symbol TSC, FSC, and SSC denote the total system cost, the system cost in period 1, and system cost in period 2, respectively.

## 5. Conclusions

In this study, an inexact optimization model was developed and examined in Shandong Province for supporting energy system planning and air pollution mitigation under uncertainty conditions. The proposed model was based on a multi-stage interval stochastic integer linear programming and the conditional value-at-risk (CVaR) measure. The optimized electricity generation, capacity expansion schemes, and economic risks were calculated and analyzed under nine defined scenarios in the paper. Results indicated that electricity generations from conventional power conversion technologies would decrease in contrast with the relative increase of clean and new power as the risk-level rises. Notably, due to the stricter environmental requirements, clean power conversion technologies were key considerations in establishing additional capacities to meet the rising social demands. Additionally, the available amounts of pollutant-mitigation and risk-aversion levels would not only cause the different power generation schemes but would also influence the total system cost. The obtained solutions could provide useful decision alternatives under different pollutant emission reduction policies and various risk aversion scenarios. Furthermore, the proposed optimization model could effectively address the uncertainties expressed as probability distributions and existing interval values, as well as could reflect the risk aversion in energy system planning problems. The solutions obtained could be valuable for supporting the adjustment or justification of air pollution mitigation management and electric power planning schemes within a complicated energy system under uncertainty conditions.

**Author Contributions:** Writing—original draft preparation, review, and editing: J.C. Writing—review and editing, and supervision: Y.X. Methodology and data curation: J.Y., Y.Y. and L.J. Supervision: C.L. All the authors contributed to the interpretation, discussion, review, and editing of pollution-emissionthe manuscript. All authors have read and agreed to the published version of the manuscript.

**Funding:** This research study was financially supported by the Guangzhou Science and Technology Project (201604020114), the Special Fund Project for Science and Technology Innovation Strategy of Guangdong Province (number 2019B121205004), and the Energy Foundation.

**Institutional Review Board Statement:** Not applicable.

**Conflicts of Interest:** The authors declare no conflict of interest.

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
