# Peer review of "A Short-Term Hybrid Energy System Robust Optimization Model for Regional Electric-Power Capacity Development Planning under Different Pollutant Control Pressures"

_sustainability, doi:10.3390/su132011341_

Round 1
Reviewer 1 Report
This work developed a short-term hybrid energy system robust optimization model for regional energy system planning and air pollution mitigation under uncertainty in Shandong province, China.
The energy system optimization model was derived from deep deconstruction of regional electric-power system in Shandong province.
Some more detailed comments are given below. I hope that if the authors will take them into account the paper will be improved.
1. A paragraph described the organization of the paper should be provided in Section 1.
2. The authors have put a lot of efforts on the energy system analysis of Shandong province in section 2, but the contribution of the paper is not very clear. The authors can comment on some of the research gap in the field and then introduce the importance or contribution of this work in Section 2.
3. It is hard to link the RIMSP method with the optimization problem (1) in Section 3.1. A simple example is needed to provide an illustration about solving this optimization problem (1) in section 3.1.
4. It is also hard to link the RIMSP method with the energy system optimization problem (2) in Section 3.2. A simple example is needed to connect the relation between problem (1) and problem (2).
5. It has almost no recent reference from the last two years (2020-2021). Recent and state-of-the-art references should be cited on the topic.
Reviewer 2 Report
The manuscript "A Short-Term Hybrid Energy System Robust Optimization Model for Regional Electric-power Capacity Development Planning under Different Pollutants Control Pressures” is interesting an relevant. I have the following comments
The novelty and research contribution of the study should be clarified and reinforced. The model is focused in the Shandong province but the research justification (last paragraph of page 2) indicates that the model is being developed for regional energy structure in China. Is the proposed model applicable to other regions of China, considering that it is being developed for one specific region? Why is this case study applicable for other regions in China? To what extend can this model be generalized?
The four steps describing the detail of the study (first paragraph page 3) should be clarified. What is the interconnection between these points. Are these points sequential or is there any interconnection on their development? I would suggest including a flow chart to indicate the process being followed and the interconnection between the research activities.
The Introduction should be improved. There is significant disconnection between the different topics discussed. I would suggest developing a narrative flow that allow readers to interconnect the diverse topics being discussed in the introduction.
Tables 1 and 2 are not clear to read. I would suggest reformatting, using two color schemes to identify diverse categories in the row of the tables. If some of the content of the tables could be converted to bar charts it may improve clarity.
The diverse scenarios discussed at the beginning of page 11 would be clearer if integrated in a table, flow chart or graph.
The description of the results in section 4 is somehow confusing. “The format the coal-fired power generation would be [1738.52, 1771.41] × 103 GWh, [1664.98, 1711.63] × 103 GWh, and [1673.93, 1715.12] × 103 GWh under S_1, S_2, and S_3; and the wind power generation would be 98.15 × 103 GWh, [137.35, 151.11] × 103 GWh, and [137.35, 145.64] × 103 GWh under S_1, S_2, and S_3, respectively.” Is used in several instances, making it hard to read the results considering the number of generating factor and parameters being described and the use of "respectively". I would suggest incorporating these results in a clearer narrative format, probably a table or graph.
The conclusion is very limited. I would suggest expanding, incorporating some of the relevant results to showcase the novelty and scientific contribution of the research. Including the limitations and future research of this study in the conclusion would be important for the conclusion.
I recommend the manuscript to be edited and revised by a native speaker. There are significant word misuse (duo instead of due in page 12, for example) and many very long sentences (for instance first sentence of section 3.3. has 67 words). Sentences should not exceed 30 – 35 words.
Reviewer 3 Report
The paper proposes an optimization model aiming to minimize the electric power system cost considering air pollution mitigation and avoiding environment risks. The model was simulated under the nine scenarios in case of the Shandong province; the obtained results are reasonable and useful for policy makers.
The following issues are recommended to improve the paper:
- Recommendation to improve the English style and grammar by a native speaker (only in a few cases).
- Solve the typing mistakes, like font size inconsistency in “However, in most energy system planning, these environmental effects and their interactions with energy development and utilization are often ambiguous or uncertain” (higher size); “The energy system optimization model model-based on”; “(i.e., 20%, 60% and 20% corresponding to…”
- For the sake of clarity, define all symbols used in equations, e.g. α, v, p, etc. in Eq. (1) and (1x). Similarly for Eq. (2) and its subsequent equations.
- (1): comment on the significance of the objective function and its terms in relation with the addressed problem.
- A Nomenclature section is strongly recommended to introduce all acronyms and symbols used in the paper. Ina such a case, the Section 3.2.1 and 3.2.2 could be removed.
- 4c is about PV power (also depicted in Fig. 3c) instead of nuclear power as stated in figure caption!
- 5 and Fig. 6: please explain why upper bound values are less than lower bound ones in several cases!
- The Conclusion section could be extended with the more relevant specific conclusions drawn from the case study of Shandong province.
Round 2
Reviewer 1 Report
The authors have carefully addressed the previous comments of the reviewer and significantly improved the manuscript.
Reviewer 2 Report
The review comments were correctly addressed in this revised version of the manuscript.
Reviewer 3 Report
No additional recommendations.